# Peeling Back the Layers: Interpreting the Storytelling of ViT

## ABSTRACT

By integrating various modules with the Visual Transformer (ViT), we facilitate a interpretation of image processing across each layer and attention head. This method allows us to explore the connections both within and across the layers, enabling a analysis of how images are processed at different layers. Conducting a analysis of the contributions from each layer and attention head, shedding light on the intricate interactions and functionalities within the model's layers. This in-depth exploration not only highlights the visual cues between layers but also examines their capacity to navigate the transition from abstract concepts to tangible objects. It unveils the model's mechanism to building an understanding of images, providing a strategy for adjusting attention heads between layers, thus enabling targeted pruning and enhancement of performance for specific tasks. Our research indicates that achieving a scalable understanding of transformer models is within reach, offering ways for the refinement and enhancement of such models.

## CCS CONCEPTS

• **Computing methodologies** → *Knowledge representation and reasoning*.

## KEYWORDS

ViT, representation, attention map, interpretability, explainability

## 1 INTRODUCTION

Since the introduction of CLIP (Contrastive Language-Image Pre-training)[26], the model has leveraged the concept of contrast to derive image representations from natural language supervision, thereby integrating image and text representations more effectively. This method transcends traditional supervision signals by harnessing the rich expressiveness of language and utilizing extensive datasets for training. Not only has this approach enhanced performance across various downstream tasks, but it has also established a robust bridge between text and images.

Inspired by this, numerous studies have sought to enhance CLIP, such as by teaching the model to focus on specific domains[31], or by attempting to increase the complexity of the CLIP model[13, 30]. This includes strategies like reconstructing masked-out, image-text aligned vision features. Other researchers have explored utilizing ViT in isolation, complemented by additional modules. This approach aims to fully exploit the semantic information acquired during the pre-training phase. Such researches, as demonstrated

*ACM MM, 2024, Melbourne, Australia*
© 2024 Copyright held by the owner/author(s). Publication rights licensed to ACM.
ACM ISBN 978-x-xxxx-xxxx-x/YY/MM
https://doi.org/10.1145/nnnnnnn.nnnnnnn

in works like [7, 16, 17], have contributed to the advancement of pre-trained vision models and large language models (LLMs) in the vision-language domain. However, these advancements largely overlook a crucial question: how exactly does the model understand the images we input during the image encoding stage?

Although some research have already been conducted to explore the processes within image encoders, Like Gandelsman[14], this work analyzes the later attention layers of CLIP-ViT and breaks down its representations into interpretable textual directions. It attributes these representations to specific attention heads and image locations, offering insights into CLIP-ViT's internal structure. Similarly, Timothée[8] have employed heatmap analyses to investigate. Their research reveals that these artifacts are outlier tokens with significantly higher norms, which manifest in transformer models under certain conditions. These tokens do not primarily carry local image information; instead, they appear to encapsulate global attributes. However, while these studies, which focus on the granularity of attention heads and image tokens, explain the interpretability of CLIP-ViT's internal mechanics, they also guide us toward a deeper question. This question emerges particularly as we navigate the transition from the micro-level (attention heads and image tokens) to the macro-level (the layers within the ViT): What specific information do the layers within the ViT pay attention to? This inquiry highlights the intricate interplay between attention mechanisms, the representation of image tokens, and the overall depth of the ViT architecture—an area that remains unexplored.

The CLIP-ViT architecture, its use of attention mechanisms allows for focusing on and integrating information from various image areas. This enables CLIP-ViT to highlight relevant features, leading to a deeper insight into the visual data. Crucially, given that CLIP-ViT can be integrated with numerous components, such as combining with the CLIP-TexT encoder to map the vision-language space, this allows for the analysis of relationships between images and text. Directly connecting a text decoder enables the conversion of captured image features into natural language descriptions, it significantly enhances the interpretability of the model's decision-making process, making it easier to trace how specific image aspects contribute to the model's output. Alternatively, appending an MLP and classification layer enables zero-shot or few-shot learning, serving as an evidence to the utility of the extracted image features.

Our initial analysis, utilizing heatmaps to examine the domains of interest within input images while discounting the impact of outlier tokens[8]. This analysis reveals that each layer of CLIP-ViT possesses a distinctive focus, engaging with different aspects of the visual field. As depicted in Figure 1, each layer of the heatmap has distinct focal points; the earlier layers familiarize with the environment, middle layers then focus on characters and details, while the upper layers distribute attention more evenly. This delineation underscores the unique contribution of each layer to the overall interpretative process. Building on this foundation, we explore converting the intricate visual analysis performed by CLIP-ViT into

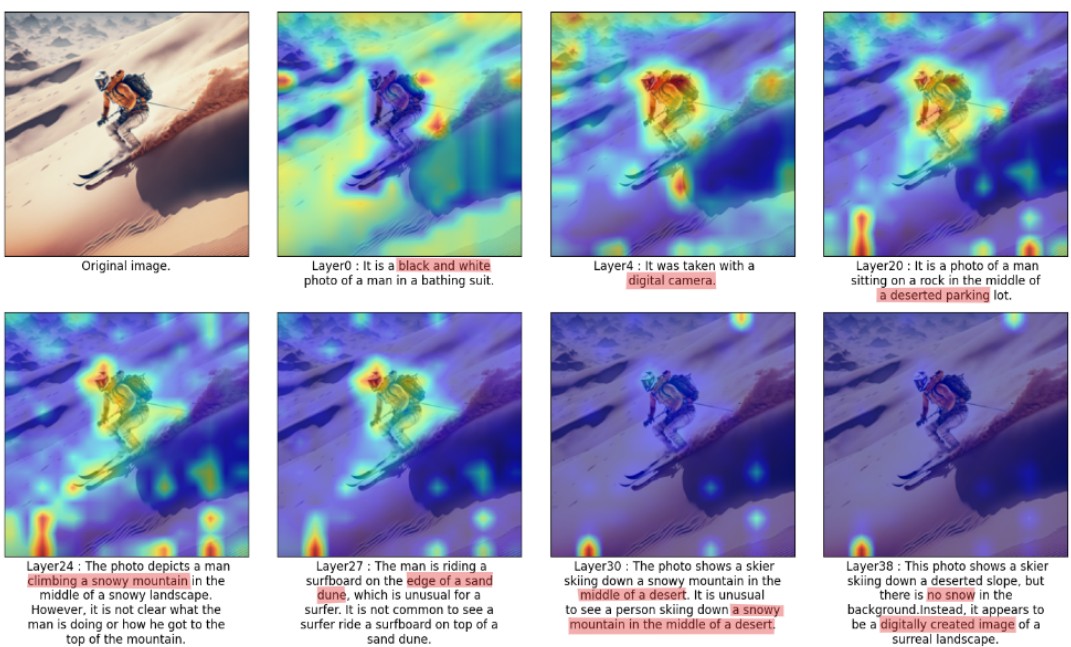

**Figure 1: The original image, heatmaps, and semantic descriptions for each layer reveal noticeable shifts in focus areas and a gradual transition in semantic expressions from abstract to concrete.**

human language by directly interfacing a decoder with these layers. This approach reveals that the generated textual descriptions encompass a diversity of information, mirroring the heterogeneity observed in visual focus. Moreover, the sequential aggregation of outputs across layers demonstrates a discernible trajectory from abstract to concrete. This progression is exemplified by the evolution from "climbing a snowy mountain" to "edge of a sand dune," culminating in the realization of "a digitally created image." Such findings convey a profound layered processing within CLIP-ViT, mapping a procedure from abstract conceptualization to precise detail, thereby enriching our comprehension of its analytical prowess.

In our exploration of the distinctive variances across successive layers within CLIP-ViT, our objective is to delineate the origins of these differences. Through attentive investigation, we discover that the variation between the initial and final layers can be attributed to the cumulative impact of the attention mechanisms deployed within the network. This reason indicates that the discrepancies noted between individual layers predominantly stem from the aggregate effect of the network's attention heads. Consequently, by modulating the configuration of attention heads across the layers, it becomes feasible to guide the model's focus towards specific facets of the visual input, thereby augmenting its sensitivity to the aspects regarded as critical. This method of modulating inter-layer attention provides a granular level of control over how CLIP-ViT processes and discerns visual information. For instance, amplifying the influence of attention heads that favor particular image attributes can enhance the model's acuity for those attributes throughout

its depth. In contrast, mitigating the influence of certain attention heads can decrease the model's susceptibility to less relevant or potentially confusing features, thus refining its concentration on the most essential elements.

We make several contributions to the field, as outlined below:
● By meticulously designing the combination of modules within the ViT architecture, we have enabled the concrete visualization of differences across layers and in natural language descriptions, facilitating subsequent analysis by humans.
● We conduct a quantitative analysis of the roles played by layers across the lower, middle, and upper layers of the ViT, including the implications of their removal. This provides new insights into and explores the black box of these models, offering a clearer understanding of their internal mechanisms.
● Through a quantitative assessment of the differences between layers, this study introduces a method for adjusting the configuration of attention heads between layers. This approach allows for the model's focus to be concentrated on specific aspects of the visual input, enhancing its analytical precision.

## 2 RELATED WORK

### 2.1 Vision model interpretability

A main stream of explainability techniques focuses on generating visual explanations by creating heatmaps. The representative examples include Grad-CAM[28], Integrated Gradients[32], LRP[3], and SHAP[19], which have been widely adopted in various domains to enhance the interpretability[5, 36]. While these heatmap-based

methods can effectively explain the relevance of specific regions in an image to the model output, they are unable to intuitively demonstrate how attributes that cannot be precisely localized to specific areas of the image (such as an object's size, shape, color, texture, etc.) influence the model's predictions. These non-spatially localizable attributes often holistically affect the model's perception and judgment of objects, but heatmap methods try to explain and visually represent their influence.

## 2.2 Intermediate representations explainability

Intermediate representations interpretability offers an approach to deciphering deep learning models. One way in this domain is feature inversion, which aims to reconstruct input images from learned features at various layers of the model[11, 15, 20]. By visualizing these reconstructions, we can reveal the visual concepts encoded by the model, demonstrating its hierarchical feature learning process from simple textures to complex semantics. Another line of research investigates individual neurons and their connections[1, 2, 23]. These studies have identified neurons with specific visual concept selectivity, revealing the model's structured information flow.

These interpretability approaches are now increasingly complemented by textual descriptions. This evolution aims to better align the model's learned features with human semantic knowledge.

Yuksekgonul et al. project model features onto a set of text-based concepts, allowing visual features to be characterized using human-interpretable semantic concepts[37]. Gandelsman's work analyzes the content focused on by the last four layers of attention heads in CLIP-ViT, revealing through visualization the visual elements each head concentrates on[14]. Goh et al. analyze the intermediate representations of the multimodal model CLIP, identifying "multimodal neurons" responsive to different visual presentations of the same theme[15]. This suggests that CLIP establishes a tight link between visual and semantic elements.

Our work distinguishes itself from these studies by our ability to pinpoint and analyze natural language interpretations for each layer within CLIP, utilizing the differences between layers to leverage the intrinsic language-image space inherent in the CLIP architecture.

## 3 DECOMPOSING ViT LAYER-BY-LAYER

This section explores how the ViT processes visual information across its layers within the CLIP framework, where ViT acts as a crucial image encoder. We begin by outlining CLIP's architecture to understand ViT's role and then delve into ViT's functionality, focusing on Multi-Head Self-Attention and Multilayer Perceptron. The analysis highlights how attention distribution across layers and heads contributes to the emergence of inter-layer differences. Building on this foundation, the differences between layers are analyzed from three perspectives:

• **Data Flow Perspective (Subsection 3.3):** By examining the flow of data through the self-attention mechanism across different layers and heads, we illustrate the process through which inter-layer differences arise.

• **Heatmap Perspective (Subsection 3.4):** Introducing the concept of average attention scores, we define a measure of difference between adjacent layers' heatmaps, visually demonstrating variations in how different layers focus on image regions.

• **Linguistic Perspective (Subsection 3.5):** By decoding the feature representations of different layers into human-interpretable language descriptions, we observe a transition from abstract to concrete descriptions with increasing layer depth, reflecting the hierarchical nature of ViT in semantic understanding.

These three perspectives essentially indicate the same phenomenon, that is, the differences between layers.

## 3.1 CLIP Architecture

**Details of CLIP**. The CLIP model maps images and texts into a shared representation space through contrastive learning, acting as a bridge between the two mediums. CLIP comprises two encoders: a Transformer-based text encoder $M_{\text{text}}$ and an image encoder $M_{\text{image}}$, which map the input text description $t$ and image $I$ into the same representation space, respectively.

**Relationship Between CLIP-ViT and CLIP-TexT**. CLIP-ViT and CLIP-TexT can be regarded as two parallel modules connected through linear projection, collaboratively constructing and showcasing a shared semantic space during the contrastive learning process. However, a examination of CLIP-ViT reveals its inherent ability to capture semantic information from images. This capability is affirmed when a linear layer is added to CLIP-ViT for classification tasks[29, 35] or when a text decoder (such as the Q-former + LLMs in BLIP series[7, 16, 17]) is attached to generate image descriptions, demonstrating its proficiency in semantic understanding.

Crucially, the semantic space manifest by CLIP-ViT does not rely on the presence of the CLIP-TexT module. In fact, CLIP-ViT, through its interaction with CLIP-TexT during the learning process, has internalized this shared semantic space. Even in the absence of CLIP-TexT, CLIP-ViT is still capable of maintaining and utilizing the knowledge of this semantic space.

## 3.2 CLIP-ViT Structure

The Vision Transformer is a model architecture designed for computer vision tasks, notably utilized as the backbone network for image representation in CLIP. For an input image $I \in \mathbb{R}^{H \times W \times 3}$, ViT first divides it into $N$ non-overlapping patches. These patches are then projected into a $d$-dimensional vector space through a linear transformation, resulting in $N$ $d$-dimensional patch vectors, which serve as the input sequence to the transformer. In addition to the patch vectors, a learnable class token $z_0^0 \in \mathbb{R}^d$ is introduced, which remains at the first position throughout the computation process of ViT and ultimately serves as the representation of the entire image. The class token $z_0^0$ and the $N$ patch vectors $z_1^0, \ldots, z_N^0$ are concatenated column-wise to form the initial state matrix $Z^0 \in \mathbb{R}^{d \times (N+1)}$.

ViT consists of $L$ identical layers, each layer featuring two submodules: Multi-Head Self-Attention (MSA) and Multilayer Perceptron (MLP). Let $Z^{l-1}$ represent the output of the $l\text{-}1_{th}$ layer. The computation within the $l_{th}$ layer unfolds as follows:

$$\hat{Z}^l = \text{MSA}^l\left(Z^{l-1}\right) + Z^{l-1} \tag{1}$$

$$Z^l = \text{MLP}^l\left(\hat{Z}^l\right) + \hat{Z}^l \tag{2}$$

Here, $\text{MSA}^l$ and $\text{MLP}^l$ denote the multi-head self-attention and multilayer perceptron at the $l_{th}$ layer, respectively. The outputs from these sub-modules are combined with their respective inputs

through residual connections, which assist in the learning process by ensuring a smooth gradient flow across the network.

Following Elhage et al.[12], the MSA output is expressed as a sum over $H$ independent attention heads and the $N$ input tokens.

$$\left[\text{MSA}^l\left(Z^{l-1}\right)\right]_{cls} = \sum_{h=1}^{H}\sum_{i=0}^{N} x_i^{l,h}, \quad x_i^{l,h} = \alpha_i^{l,h}\left(z_i^{l-1}W_V^{l,h}\right) \quad (3)$$

where: $H$ represents the number of attention heads, $N$ signifies the length of the input sequence, $x_i^{l,h}$ indicates the contribution of the $i_{th}$ input token to the class token at the $l_{th}$ layer and $h_{th}$ attention head, $\alpha_i^{l,h}$ denotes the attention weight the class token assigns to the $i_{th}$ token at the $l_{th}$ layer and $h_{th}$ attention head, with $\sum_{i=0}^{N} \alpha_i^{l,h} = 1$. $W_V^{l,h} \in \mathbb{R}^{d \times d_h}$ is the value matrix of the $h_{th}$ attention head at the $l_{th}$ layer, transforming $d$-dimensional input tokens into $d_h$-dimensional value vectors ($d_h = d/H$). $z_i^{l-1} \in \mathbb{R}^d$ represents the $i_{th}$ output token of the $l-1$ layer. This decomposition clearly illustrates how the output of the multi-head self-attention is assembled from the contributions of each layer, attention head, and input token. Such a decomposition aids in a deeper understanding of ViT's internal mechanics and lays the groundwork for subsequent interpretability analysis.

### 3.3 Data Flow Perspective

Based on Equations 1 and 2, we can represent the information difference between two adjacent layers, the information difference between adjacent layers, $\Delta Z^l$, can be expressed as:

$$\Delta Z^l = Z^l - Z^{l-1} = \text{MLP}^l\left(\hat{Z}^l\right) + \text{MSA}^l\left(Z^{l-1}\right) \quad (4)$$

Expanding $\text{MSA}^l\left(Z^{l-1}\right)$ at the class token position into the form of attention heads as Equation 3, we obtain:

$$\Delta Z_{cls}^l = \left[\text{MLP}^l\left(\hat{Z}^l\right)\right]_{cls} + \sum_{h=1}^{H}\sum_{i=0}^{N} x_i^{l,h} \quad (5)$$

Therefore, the information difference between two adjacent layers can be represented as the sum of the outputs from the MLP and the multi-head self-attention mechanism.

However, this concept can be extended to all positions in the sequence. For the $j_{th}$ output token at layer $l$, the information difference $\Delta Z_j^l$ relative to the corresponding position in the previous layer can be fully represented as:

$$\Delta Z_j^l = \left[\text{MLP}^l\left(\hat{Z}^l\right)\right]_j + \sum_{h=1}^{H}\sum_{i=0}^{N} \alpha_{i,j}^{l,h}\left(z_i^{l-1}W_V^{l,h}\right) \quad (6)$$

where, $\left[\text{MLP}^l\left(\hat{Z}^l\right)\right]_j$ denotes the output of the MLP at position $j$ in layer $l$. $H$ indicates the number of attention heads. $N$ represents the length of the input sequence. $\alpha_{i,j}^{l,h}$ is the attention weight the $j_{th}$ output token at layer $l$, head $h$, assigns to the $i_{th}$ input token, satisfying $\sum_{i=0}^{N} \alpha_{i,j}^{l,h} = 1$. $W_V^{l,h} \in \mathbb{R}^{d \times d_h}$ is the value matrix of the $h_{th}$ attention head at layer $l$, transforming $d$-dimensional input tokens into $d_h$-dimensional value vectors ($d_h = d/H$). $z_i^{l-1} \in \mathbb{R}^d$ represents the $i_{th}$ output token of layer $l-1$.

This expression demonstrates how the information difference at the $i_{th}$ position of layer $l$ is comprised of the MLP output and the weighted contributions from all attention heads at that position. Through this decomposition, we can gain a deeper understanding of the dynamics of spatial information transmission and updating within the ViT model, and how different attention heads capture and integrate information at various positions.

### 3.4 Heatmap Perspective

At a macro level, we can observe the differences between layers by comparing the attention heatmaps across different layers. For instance, the attention heatmaps of layers, as illustrated in Figure 1, show significant differences, reflecting their distinct focuses in processing visual information.

This variability can be captured using equations 5 and 6, the term $\sum_{h=1}^{H}\sum_{i=0}^{N} x_i^{l,h}$ represents the sum of contributions from all attention heads at the CLS position in layer $l$. The difference in this term directly manifests in the attention heatmaps.

To better describe the variability observed in the heatmaps, we use $A^l$ to represent the average attention score at layer $l$:

$$A^l = \frac{1}{H}\sum_{h=1}^{H}\sum_{i=1}^{N} \alpha_i^{l,h} \quad (7)$$

where $\alpha_i^{l,h}$ denotes the attention score assigned by the $h_{th}$ attention head at layer $l$ to the $i_{th}$ input token. With $A^l$, we can define the difference in average attention scores between adjacent layers, $\Delta A^l$:

$$\Delta Z^l \equiv \Delta A^l = A^l - A^{l-1} \quad (8)$$

$\Delta A^l$ directly reflects the variability between layers $l$ and $l-1$ on the attention heatmaps. The greater the absolute value of $\Delta A^l$, the more pronounced the difference in attention distribution between the two layers; conversely, the smaller the absolute value of $\Delta A^l$, the more similar the attention distribution between the two layers.

### 3.5 Linguistic Perspective

By decoding $Z^l$, we can convert the visual analysis process of CLIP-ViT into human understandable language descriptions, thus allowing us to more intuitively observe the differences between layers. As illustrated in Figure 1, with increasing layer depth, the generated text descriptions evolve from abstract to concrete. We introduce a decoder $D$ to denote the process of converting $Z^l$ into natural language descriptions, where $D : \mathbb{R}^{d \times (N+1)} \rightarrow \mathcal{S}$, with $\mathbb{R}^{d \times (N+1)}$ representing the space of $Z^l$ and $\mathcal{S}$ denoting the space composed of natural language descriptions. $D$ is a Transformer-based autoregressive language model. The output representation $Z^l$ from layer $l$ is first mapped to an embedding layer, from which $D$ subsequently generates the corresponding natural language description.

This allows us to extend the concept of $\Delta Z^l$ into the linguistic domain. Specifically, we can define $\Delta S^l$ to represent the difference in language descriptions between layer $l$ and layer $l-1$:

$$\Delta Z^l \equiv \Delta S^l = \text{Metric}(D(Z^l), D(Z^{l-1})) \quad (9)$$

Here, Metric denotes a chosen standard of measurement (such as BLEU[25], ROUGE[18], BERTScore[38], CIDEr[33], or using GPT-4[24] to score) to assess the difference between two descriptions.

As the layer number $l$ increases, according to the Equations 2, $Z^l$ is continuously updated and enriched, integrating information from all preceding layers. Descriptions from earlier layers might be more abstract and general, while those from later layers become more specific and detailed. When we decode $Z^l$, the generated text descriptions will reflect these changes in hierarchy and detail. This transformation aligns with the evolution of $Z^l$, providing a deeper insight into the model's internal mechanisms through the lens of human-interpretable language.

## 4 EXPERIMENT

In our experimental section, we start from a classic conclusion that in ViT, the lower layers are responsible for processing low-level features, focusing on local texture details, the middle layers introduce concepts of object shapes and handle abstract features, while the upper layers are tasked with processing high-level semantics and capturing more global information[10, 22, 27]. We employ the three perspectives —Data Flow, Heatmap, and Linguistic Perspectives—to quantitatively analyze these aspects in a combined manner.

### 4.1 Experimental Information

**Dataset Selection** To thoroughly validate the effectiveness of our experiments, we conduct tests across multiple datasets. These include WHOOPS[4], which features commonsense-defying images for a variety of reasons, including deviations from expected social norms and everyday knowledge. We also include the classic ImageNet dataset[9] in our experiments; however, due to its size, we only conduct experiments on the validation set.

**Model Selection** Our study primarily focuses on EVA-CLIP-ViT[13, 30], a model with a 40-layer ViT architecture. This choice of architecture is deliberate, as the 40-layer configuration provides a more granular insight into the hierarchical processing of visual information compared to the short layer variant. This depth enables us to more thoroughly investigate the transition from low-level texture details to high-level semantic understanding, aligning with our goal to analyze the meticulous ways in which ViT architectures interpret and analyze visual data. In the section on the Linguistic Perspective, we employ a decoder using instructBlip[7], which appends a Q-former behind its own ViT to obtain image query tokens. This setup is further extend with a flan-t5-xl[6] for decoding. Our experiments are conducted entirely in a zero-shot manner, without any fine-tuning or training. The GPU use for our experiments is equipped with 80 GB of VRAM, specifically the A800 model.

### 4.2 Lower layers: The Texture Tailors

***Experiment Setting.*** In this experiment, our aim is to delve into the details that the lower layers of the ViT model (specifically layers 0-10 and 11-20) focus on, as well as the main features processed by these layers. From prior research, we understand that the lower layers of the ViT model tend to capture texture details in images. However, the specifics of these texture features and how they vary between different layers require further investigation.

To determine whether the features extracted by the lower layers exhibit universality and shareability, we conduct experiments on the Whoops dataset as follows: First, we extract the output features of the ViT model's layers 0-10 and 11-20 for each image in the dataset.

Then, we average the output features across the entire dataset to obtain an average feature vector for each layer. We consider this average vector as a representation of the shared lower-level features extracted by the corresponding layer. The Whoops dataset is chosen for our experiments because its task is to describe strange or unusual aspects of images. This demands from the model an understanding of details, commonsense, and context, where any difference directly reflects in the outcomes.

By analyzing the image descriptions generated by the model, i.e., through the perspectives of Data Flow and Linguistic Perspectives, we can understand the role of lower-level features in capturing anomalies in images, and thereby infer the types of texture details these features represent. To validate the effectiveness and generalizability of these average feature vectors, we apply them to a downstream image description generation task. Specifically, we replace the input to the Multi-Head Self-Attention (MSA) in Equation 1 with these average feature vectors, so the model's attention mechanism operates solely on these average vectors. This allows us to evaluate the effectiveness of generating image descriptions using only shared lower-level features.

***Experiment Result.*** Figure 2 illustrates the changes in BERTScore values between the model's output and the original output after averaging the features of different layers in the ViT model. The x-axis represents the layer number (0-20), and the y-axis shows the BERTScore value after averaging for each layer.

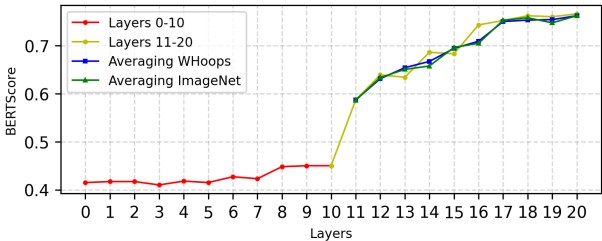

**Figure 2: The graph averages results from the bottom and middle-lower layers.**

The results indicate that averaging any 0-10 layers significantly impacts the model's output, as shown by the red curve in the graph. Upon further check of the averaged natural language description outputs, we observe that the content becomes disordered, the output cases will be shown in Appendix A.

However, averaging any layer within the 11-20 range seems not to have a significant impact on the model's output, as indicated by the yellow curve. Further experiments averaging all layers between 11-20 simultaneously also show no significant effect, as depicted by the blue curve. To further explore the role of layers 11-20, we randomly select 500 images from the ImageNet dataset, extract their average vectors for layers 11-20, and then replace the corresponding layers in the WHOOPS dataset with these average vectors. Surprisingly, this replacement still does not significantly affect the model's output, as shown by the green curve.

Does this mean that layers 11-20 are unimportant to the model's decision-making process? To answer this question, we attempt to "confuse" these layers by replacing the residual connections with

average vectors in Equation 1. We find that such "confusion," even if it occurs in just one layer, leads to a sharp decline in model performance. Additionally, in the natural language descriptions, we observe mentions of "black and white photos".

***Experiment analysis.*** In this section, we address three questions:

**(1) Impact of Averaging Layers 0-10:** In the ViT model, layers 0-10 are primarily responsible for extracting low-level image features. These features lay the foundation for subsequent high-level feature extraction and semantic understanding. Averaging these layers can break these crucial low-level features, thereby affecting feature extraction in subsequent layers and the final output. The model struggles to identify and make sense of the modified basic features.

**(2) Lack of Impact from Averaging Layers 11-20:** The middle to lower layers (11-20) of ViT process more abstract and complex features, such as parts of objects, entire objects, and ultimately achieving global scene understanding. By the time visual information reaches these layers, it has been abstracted to a level where averaging may not avoid the essential features the model relies on for high-level reasoning. The absence of significant impact from replacing averaged vectors from layers 11-20 in the WHOOPS dataset onto the model suggests that these layers are robust against averaging. Even downplayed, the averaged vectors might still contain enough semantic information for the model to generate accurate outputs. This resilience underscores the model's ability to extract meaningful information from even compromised abstract features.

**(3) Drastic Performance Decline from "confusion" Layers:** Directly "confusing" layers 11-20, by replacing residual connections with averaged vectors, leads to a drastic decline in model performance. This finding may seem contradictory to previous results, where averaging layers 11-20 did not affect model outputs. Altering the residual connections fundamentally changes how information is propagated through the model, possibly leading to a failure in correctly integrating and interpreting visual information and causing substantial performance decreases. Experiments reveal that when residual connections are replaced with average vectors, descriptions such as "black and white photos" emerge, suggesting layers 11-20 might be closely related to color information expression.

### 4.3 Upper layers: The Semantic Sculptors

***Experiment Setting.*** In investigating the impact of the ViT's top layers (primarily layers 31-38) on model performance, we adopt an approach that combines Data Flow and Linguistic Perspectives. Similar to previous experiments, we compare the model's output at different layers with the golden label on the WHOOPS dataset. Unlike experiments with lower layers, here, the focus is more on the influence of high-level abstract semantics on the final golden label rather than comparing it with the final layer's output, to assess each layer's contribution to the overall output quality. Figure 3 presents the experimental results, with the horizontal axis representing the layer numbers and the vertical axis showing the average CIDEr calculated across the entire dataset.

***Experiment Result.*** The results indicate that as the number of layers increases, the CIDEr gradually improves. However, starting from layer 35, the increase in CIDEr begins to be stable.

***Experiment Analysis.*** In this section, we address two questions:

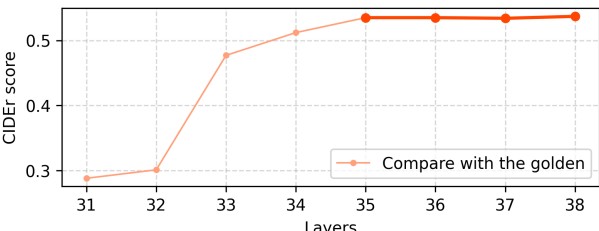

**Figure 3: The CIDEr as it changes from Layers 31 to 38.**

**(1) The result improvement tend to plateau after layer 35:** The plateauing after layer 35 can be attributed to the model reaching a point of diminishing returns in terms of extracting and integrating additional semantic information from the visual input. The top layers of the ViT model are designed to capture high-level semantic features and global contextual information. By layer 35, the model has likely already captured the most significant semantic features relevant to the task, and additional layers contribute marginally smaller improvements to the final output quality.

**(2) Explanations from the perspectives of Data Flow and Linguistic Perspectives:** • Data Flow Perspective. By the time information flows through to the upper layers (beyond layer 35), it's already highly abstracted, representing complex scene configurations and relationships among objects within a global context. Further layers add only marginal semantic differentiation or refinement to this already comprehensive visual understanding, thus offering minimal additional benefit to the task of semantic interpretation from a visual standpoint. • Linguistic Perspective. The essence of the plateau from a linguistic perspective is that the visual semantics necessary for text generation have reached a saturation point; the model's linguistic output cannot be significantly improved without additional external input or a more detailed processing mechanism that can extract further subtleties from the visual data.

### 4.4 Middle layers: The Concept Weavers

***Experiment Setting.*** In exploring the mid-layers (20-30) of the ViT model, we integrate observations from sections 4.2 and 4.3, discovering these layers play a crucial role in generating accurate descriptions. Analyzing from the Linguistic Perspective, we observe a transition from "abstract to detail" in the upper mid-layers.

For instance, the model refines "a large animal" to "a giraffe," specifies "a white statue" as "a snowman," and identifies "plants" as "cacti." This transition from abstract descriptions to concrete entities highlights the mid-layers' significant role in semantic expression and detail capture, the cases will be shown in Appendix B. To further validate this finding, we conduct experiments on the WHOOPS dataset, particularly examining the model's performance on questions requiring high detail.

Overall, the research process of this subsection is as follows: We first analyze the connection between this abstract-to-detail transition and model performance from both the Heatmap Perspective and the Linguistic Perspective. Then, we carefully design prompts and used GPT-4V(ision) to score the semantic descriptions of these layers, objectively evaluating the "abstract to detail" phenomenon

we identified. After experimental validation, we further explore how such macro-level differences emergence. We focus on the attention heads within the same layer. By analyzing the data flow, we discover that certain attention heads play a key role in capturing and conveying detail information. Based on this finding, we attempt to prune some irrelevant heads, and the result showed that doing so could enhance the model's attention to details, further validating the effectiveness of the Data Flow Perspective.

***Heatmap and Linguistic.*** As illustrated in Figure 1, the heatmaps and semantic descriptions work synergistically. In the appendix B, we provide more samples of heatmaps and semantic descriptions. These comparison images illustrate the attention distribution across different layers and their corresponding semantic descriptions, visually presenting the transition from abstract to detail. On the heatmaps, there is a process from focusing on general areas to specific detail parts , in terms of semantic descriptions, is reflected as a shift from general descriptions to the capture of details.

***GPT-4V(ision) Evaluating.*** In exploring semantic changes in the middle layers, we choose to use GPT-4 as an objective evaluation tool, based on two important considerations: Firstly, GPT-4 has been shown to significantly overlap with human evaluators across many tasks, providing us with reliable assessment results. Secondly, there is currently no tool as efficient and comprehensive as GPT-4 in dealing with entities and their relationships. For example, understanding the transition from "a white statue" to "a snowman," requires a rich background in semantic knowledge and the integration of image information for accurate judgment.

To fully leverage the GPT-4V, we carefully design the prompts for the evaluation task. As shown in Figure 4, we construct a structured prompt to guide GPT-4V in scoring the descriptions from different layers from a semantic perspective. This prompt clearly defines the criteria and objectives of the evaluation, asking GPT-4V to determine whether the descriptions exhibit a progression from abstract to concrete and to assign a corresponding score.

> Please evaluate the changes between different layers of the given image description and determine if the description becomes more concrete according to the following criteria:
> **Number of Entities:** The count of specific entities mentioned in the description, such as objects, people, scenes, etc. **Specificity of Entities:** Considering factors such as entity attributes and detailed descriptions, the average level of specificity of entities in the description. **Information Content of the Description:** The amount of useful information contained in the description, considering the richness and detail of the description, and whether it is more specific than the previous layer (the first input is default scored as 3). **Correspondence Between Description and Image:** The relevance and degree of match between the description and the image content.
> For each criterion, please score according to the following levels:
> 1 point: Few/Poor. 3 points: Medium/Average. 5 points: Many/Good.
> {LAYERS_LIST}

**Figure 4: The above depicts the prompt input for GPT-4V(ision), where {LAYERS_LIST} serve as placeholders, which will be provided during actual implementation.**

This approach enables us to objectively quantify the trend of semantic changes in the middle layers, thereby validating the observed phenomenon of a transition from "abstract to concrete." The specific results are illustrated in Figure 5, where the vertical axis represents the scores by GPT-4V, and the horizontal axis

shows the changes in layers. By using GPT-4V to score semantic descriptions across different layers, we observe an upward trend, indicating that the semantic descriptions indeed transition from abstract to concrete. Furthermore, the results show a significant semantic shift particularly between layers 24-28. More examples will be provided in the appendix C. The scoring results from GPT-4V provide us with objective, quantifiable evidence that supports the phenomena.

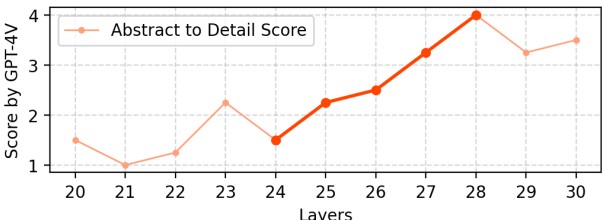

**Figure 5: Abstract to concrete score as evaluated by GPT-4V.**

***Specific Attention Head.*** To further explore the transition phenomenon from "abstract to concrete", we approach from the perspective of data flow, focusing on individual attention heads or combinations. By designing masking matrices, we selectively block the influence of certain attention heads (details described in the appendix D), thus precisely identifying the role of each head or head combination in semantic expression and detail capture. This decomposition method not only aids in understanding the information processing mechanisms within layers but also offers new insights into the study of model interpretability.

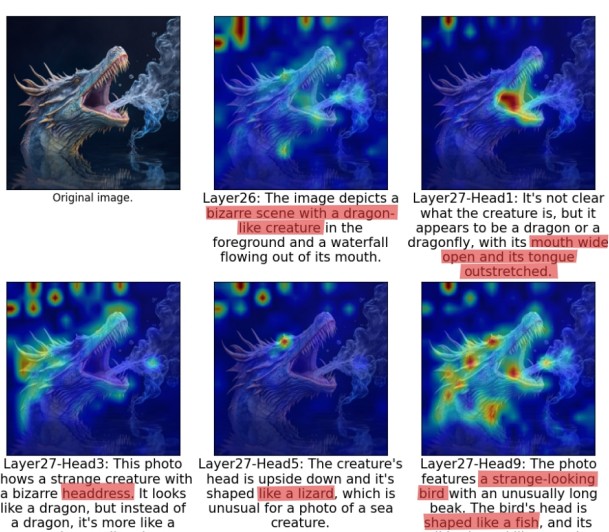

**Figure 6: The original image, Layer 26 outputs, and attention heads of Layer 27 demonstrate a correlation between heatmaps and textual descriptions, linked to specific heads.**

In Figure 6, we present heatmaps for individual attention heads and their corresponding semantic expressions. These visual results

provide rich information, revealing the diversity and complementarity of different attention heads in semantic expression and detail capture. From the heatmaps, different attention heads exhibit varying focus patterns. Some heads may focus more on the overall structure and layout of the image, while others concentrate on specific local areas and details. Furthermore, we observ significant differences in the semantic expressions generated by different attention heads. Some heads produce more abstract and generalized descriptions, whereas others generate more detailed descriptions.

These experimental results lead us to pay attention to whether we could enhance the model's focus on desired details for specific tasks through attention head pruning. Such targeted pruning offers a method for fine-tuning the model's attention mechanism, which could lead to improved performance.

**Attention Pruning.** In this part, we discuss the method of pruning attention heads based on a limited samples. Specifically, we select 100 samples and conduct the pruning operation following the procedure outlined in Algorithm 1. After completing the pruning, we evaluate two key metrics using the remaining samples to assess the effects of the pruning, with the results present in Table 1.

**Table 1: Evaluation metrics for Attention Pruning.**

|  | CIDEr ↑ | Convergence Layers ↓ |
| --- | --- | --- |
| BLIP2 FlanT5-XXL | 1.2 | 35 |
| + Attention Pruning | 1.24 | 34 |

---

**Algorithm 1** Attention Pruning

---

1: **Input:** Dataset Dataset, number of attention heads $n$ in the current layer, Decoder $\mathcal{D}$, attention head coefficients $\alpha_{i,j}^h$
2: **Output:** Adjusted attention head coefficients
3:
4: **procedure** PRUNEATTENTIONHEADS
5:     Randomly select $m$ data from Dataset to form sample set $S$
6:     **for** each data point $s$ in $S$ **do**
7:         Decode the combined output $\mathbf{Z}^s$ using $\mathcal{D}$ to get $\hat{Z}^s$
8:         **for** each attention head $h$ from 1 to $n$ **do**
9:             Extract and decode the output of head $h$, $\mathbf{Z}_h^s$, using $\mathcal{D}$
10:            Compute the BERTScore $Score_h^s = BERTScore(\mathcal{D}(\mathbf{Z}_h^s), \hat{Z}^s)$
11:         **end for**
12:     **end for**
13:     Calculate the average BERTScore for each head $h$, get $Score_h^{avg}$
14:     Compute softmax for importance scaling: $\alpha_h = \text{Softmax}(Score_h^{avg})$
15:     Adjust coefficients: $\alpha_{i,j}^h \leftarrow \alpha_{i,j}^h \cdot \alpha_h$
16:     Prune attention heads with low importance scores
17: **end procedure**

---

Firstly, we examine the convergence speed of the model after pruning. By comparing the average number of layers required for the model to converge before and after pruning, we find that the pruned model converges more quickly. The pruning process, which removes redundant or less important attention heads, makes the model more streamlined and efficient, thus accelerating the convergence process.

Secondly, we evaluate the performance of the pruned model on the dataset. By comparing the model's output with the golden label, we calculate a comparative metrics. Fortunately, the pruned model achieves better results on this metrics. This indicates that the streamlined model performs better than the original model across the entire dataset and contains fewer model parameters.

**Experiment Analysis.** In this section, we address two questions: **(1) Transition from Abstract to Concrete:** Firstly, the transition observed is related to the hierarchical nature of visual information processing, consistent with many previous research. Secondly, the middle layers serve a pivotal role in bridging shallow local features with deep semantic representations. Compared to the bottom layers, which have a limited receptive field, and the top layers, which may be overly coarse, the middle layers achieve a better balance. **(2) Effectiveness of Attention Pruning:** Firstly, not all attention heads are necessary. Different heads focus on different aspects; some concentrate on crucial features while others may capture irrelevant information. Secondly, pruning helps the model focus more on key details. Retaining well-performing heads and suppressing or removing underperforming ones acts as a form of "soft guidance." Additionally, pruning can mitigate overfitting to some extent, thereby enhancing its generalization capabilities.

## 5 CONCLUSION

The Vision Transformer (ViT) serves as a backbone framework that merits detailed exploration, particularly by analyzing its layer-by-layer processing within the CLIP architecture. We have meticulously examined how layers and their corresponding attention heads contribute to image understanding and revealing variations.

The application of a combination of data flow, heatmap, and linguistic perspectives has facilitated a comprehensive understanding of ViT's functionality, offering insights into the complex dynamics of the model. Our findings demonstrate that by modulating attention heads and their configurations, one can adjust the model's focus and enhance its interpretability. This ability to guide layer-specific processing emphasizes the potential of ViT models in applications requiring precise image analysis and interpretation.

These discoveries provide new manners for understanding and improving ViT as well as other similar transformer models, enhancing their applicability across various domains.

## 6 FUTURE WORK AND LIMITATIONS

In future work, we plan to delve deeper into the intrinsic connections between layers in the ViT model. There is a need to further explore the mechanisms of cooperation between attention heads, the methods of interaction and collaboration between heads remain to be explored. Additionally, the robustness of the ViT model across different layers has not yet been conclusively determined, such as noise, occlusions, and adversarial attacks at various levels.

We recognize some limitations in this study. Observations suggest that attention heads on certain layers may not directly influence the output of those layers but rather have a more significant impact on subsequent layers[21, 34]. This phenomenon is closely related to attention pruning techniques. Although our small-sample pruning experiments validate the effectiveness of this method, this pruning strategy might only find locally optimal solutions between adjacent layers. Therefore, we emphasize that when applying attention pruning techniques, it should be tailored to specific tasks and validated through small-sample data. This approach can ensure model performance while reducing computational costs.

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
