# OpenReview forum: "Peeling Back the Layers: Interpreting the Storytelling of ViT"
_acmmm.org/ACMMM/2024/Conference — MM2024 Oral_

### Official Review · Reviewer_KJS9 · 2024-05-22

**Rating:** 4
**Confidence:** 3

**Summary:**

The paper explores the internal mechanics of the Visual Transformer (ViT) architecture, specifically focusing on its approach to interpreting and processing visual information during the image encoding phase.The authors conduct a comprehensive analysis by integrating various modules, dissecting each layer and attention head within the ViT to unveil how the model progresses from abstract concepts to recognizable objects across different layers. The research not only underscores the visual interactions among layers but also examines how manipulating attention heads can enhance model performance for specific tasks.

**Strengths:**

[S1] This work gives a very detailed background of the study so that people not in the field can quickly understand it.
[S2] The authors validate the effectiveness of their theoretical analysis through extensive experiments across datasets, and enhance the persuasiveness of their findings by quantitatively demonstrating the specific contributions of each layer in ViT to performance.
[S3] The author puts forward the direction of future work and the limitations of the current research at the end of the article, which will help subsequent researchers continue to explore and expand the current work.

**Limitations:**

L1: While the paper excels in theoretical analysis and experimental design,it appears to lack groundbreaking innovation. It is more of an interpretive analysis of the existing ViT model rather than proposing new model architectures or significant advancements in current techniques. Specifically, although the paper introduces the Attention Pruning method, the performance improvement it brings is quite limited and does not demonstrate strong persuasiveness.
L2: The spelling of some model names does not satisfy writing conventions, such as InstructBLIP.
L3: The attention head pruning experiments mentioned in this paper were conducted based on small samples. This may mean that the pruning strategy may not have been validated on a wider range of data and its generalization ability needs further study.

**Suitability:**

2

---

### Official Review · Reviewer_hm1X · 2024-05-24

**Rating:** 4
**Confidence:** 2

**Summary:**

This paper proposes a method to explore the explainability of image processing in Vision Transformers. It includes a detailed investigation of the different functions of specific attention heads or layers within the Vision Transformer, enabling certain specific tasks.

**Strengths:**

Comprehensive experiments and evaluation metrics support the validity of the experiments.

**Limitations:**

I understand that this work is primarily exploratory. It would be beneficial if the author could demonstrate how some downstream tasks can directly or indirectly benefit from the observations and conclusions of this work.

**Suitability:**

3

---

### Official Review · Reviewer_Nbjy · 2024-05-25

**Rating:** 4
**Confidence:** 3

**Summary:**

The paper breaks-open the ViT architecture layer-by-layer and presents insights on what the role of each of the layers is.

**Strengths:**

It is an overall interesting direction where the authors are focussing on understanding the details of what exactly is happening to an image in ViT encoding process.
The analysis done is exhaustive and expands across multiple perspectives.

**Limitations:**

I am not convinced with the potential applications these insights might lead to. If the authors can present advantages of using their insights for any downstream task, it would make the paper stronger and complete.
Q - do the general observations extend to other architectures as well?

**Suitability:**

2

---

### Meta-Review · Area_Chair_32Kp · 2024-07-03

**Recommendation:** Accept (Oral)
**Confidence:** 5

**Metareview:**

The paper explores the internal mechanics of the VIT architecture, specifically focusing on its approach to interpreting and processing visual information during the image encoding phase. The analysis is exhaustive and spans multiple perspectives. Given its significance, all reviewers believe that this paper should be accepted by MM.